# A Modulated Approach for Improving MFSK RADARS to Resolve Mutual Interference on Autonomous Vehicles (AVs)

**DOI:** 10.3390/s23167192

**Published:** 2023-08-15

**Authors:** Jonathan Duke, Eli Neville, Jorge Vargas

**Affiliations:** Department of Engineering Technology, Middle Tennessee State University, Murfreesboro, TN 37132, USA; ejn2p@mtmail.mtsu.edu

**Keywords:** frequency modulation, frequency shift keying, mutual interference

## Abstract

This paper proposes a novel automotive radar waveform involving the theory behind M-ary frequency shift key (MFSK) radar systems. Along with the MFSK theory, coding schemes are studied to provide a solution to mutual interference. The proposed MFSK waveform consists of frequency increments throughout the range of 76 GHz to 81 GHz with a step value of 1 GHz. Instead of stepping with a fixed frequency, a triangular chirp sequence allows for static and moving objects to be detected. Therefore, automotive radars will improve Doppler estimation and simultaneous range of various targets. In this paper, a binary coding scheme and a combined transform coding scheme used for radar waveform correlation are evaluated in order to provide unique signals. AVs have to perform in an environment with a high number of signals being sent through the automotive radar frequency band. Efficient coding methods are required to increase the number of signals that are generated. An evaluation method and experimental data of modulated frequencies as well as a comparison with other frequency method systems are presented.

## 1. Introduction

In recent years, engineers and data analysts have been searching for ideal methods to collect data from multiple driving scenarios. Economical self-navigation can be achieved by using radar sensors rather than cameras for data acquisition. Radar sensors will acquire data to improve simultaneous range and Doppler estimation, both of which are crucial for safe navigation. Advanced driver-assistance system (ADAS) has laid out definitions for autonomous driving in six distinct stages as shown in Figure 1 to mark progress towards AV sensor-based driving functions and high-definition maps once vehicles are performing all driving functions [1,2].

Mutual interference is a major issue that prevents AVs from using radar data during periods of high traffic. This issue occurs when two devices emit the same frequency, causing each sensor to report unreliable data. Furthermore, if AVs shared radar sensor data, reciprocal interference might be avoided. Due to the mass production of most AV radar sensors, they tend to work on the same wavelengths as other AV physical devices, such as cameras, lidars, and ultrasonic sensors. An emitter on one vehicle transmits data to a receiver on another vehicle, rendering the sensor’s data worthless and potentially leading to unexpected driving decisions. Any form of sensor that produces and receives a frequency might cause mutual interference. The likelihood of receiving reciprocal interference grows as the range of any sensor increases in relation to the frequency’s emitted region. Mutual interference is more common in radars than ultrasonic devices, as they operate in a narrower regions of space [3]. Figure 2 shows an example of a common scenario that can cause mutual interference.

Many applications benefit from modulating the frequency of a sinusoidal wave. Frequency modulation (FM) was originally utilized in radio wave transmission in the 1930s [4]. Using FM instead of amplitude modulation (AM) allows radio waves to be broadcast within a greater bandwidth. The sound resolution improves while increasing interference susceptibility. Frequency modulation is a useful approach for identifying faults in AV object detection. A frequency-modulated continuous-wave (FMCW) radar may identify several objects by using Fourier transforms for frequency waveform identification, although these mathematical data currently have mutual interference constraints [5].

## 2. AV RADAR Methods

FMCW RADAR systems are built on the foundation of frequency modulation. Along with modulating frequency, FMCW is a continuous-wave RADAR which enables calculations for range by comparing transmit and receive frequencies. There are many different modulation schemes pertaining to frequency modulation, with a common modulation scheme being linear with an increasing frequency. In Figure 3, a linear FMCW waveform can be shown, modulating frequency over time. FMCW RADARs tend to modulate linearly but can be designed to operate differently. Since automotive radars can only operate within specified bands of frequencies, FMCW RADARs linearly sweep an entire frequency band. A FMCW signal will sweep from the lower end of the band to the higher end of the band and vice versa. This sweep can be in two directions, up or down. In addition, Linear FMCW waveforms can produce sawtooth or triangular patterns [5].

Linear FMCW RADARs can be mathematically represented in Equation (Equation 1):(1)$$F_{linear}=mt+f_{s};\;f_{s}=76\;\mathrm{GHz}$$
where *m* is the slope, *t* is the x-intercept, *f${}_{s}$* is the y-intercept in terms of giga Hertz (GHz) for frequency. The linear sweep allows radars to calculate the radial velocity for a given object.

### 2.1. FMCW RADAR Calculations

A Fourier transform compares samples from a signal that has propagated through space or time. When using data collected from sensors, sampling is critical in the case of AV RADARs. A fast Fourier transform (FFT) is often used by AVs to quickly calculate values from sensor signals. Using a FFT of a received signal, an AV can determine the object location based on how the signal has changed throughout propagation. The accuracy of direction and distance values of a RADAR are directly proportional to the number of receiving antennas. Using an FFT to identify the phase difference between multiple received signals, the distance can be estimated. To accomplish this, the continuous-wave RADAR system must provide IQ “in-phase/quadrature-phase” signals, which are two orthogonal sinusoidal waves [6]. Comparing the amplitude of the IQ signals across two different receiving antennas allows for distance to be calculated in Equation (Equation 2):(2)Anobjectdistance=dSinθ
where *d* is the distance between two receiving antennas and θ is the phase angle of the received signal.

As the number of received antennas on a radar module increases, the RADARs angular resolution increases as well. For continuous-wave RADARs, a higher angular resolution enables a more accurate distance value to be calculated. The Doppler and time delay can be estimated after performing a FFT of the received signal. Figure 4 shows the difference in emitted (in red) and received (in green) signals along the x-axis for the time delay (*Δt*) and the y-axis for frequency (*Δf*). Doppler frequency data (*f${}_{D}$*) can be calculated similarly on the y-axis between emitted and received signals.

Since time delay and Doppler data are vectors on their own axes, they can be shown alongside one another to aid with object tracking. When AV RADARs produce IQ signals, two plotted vectors containing time delay and Doppler data intersect with one other, resulting in the true time delay and Doppler data for a specific object. Increasing the number of time delay and Doppler data vectors enable more accurate readings to be measured.

#### Ghost Detections

As seen in Figure 5, each red circle is an object’s true location. A RADAR system that is detecting two objects can be visually seen as any two of the red circles with the arrows pointing towards them. The other two red circles are known as ghost detections. The vectors shown are derived from the time and Doppler delay magnitudes of the received signal from an object. For FMCW RADARs, ghost detections occur when multiple objects are being tracked. FMCW RADARs are limited due to their ability to only output a linear modulation pattern. Since FMCW RADARs output a triangular wave, the time delay and Doppler values are the same. Therefore, each circle in Figure 5 has two vectors crossing through them. Each point where a vector overlaps creates an indication of an object’s true location. When tracking multiple objects, these vectors start to overlap and provide false locations. A constant false alarm rate (CFAR) is commonly adjusted to allow RADARs to perform while also picking up false readings. Increasing the number of time delay or Doppler vectors enables more vectors to be used in the elimination of ghost detections [7,8].

### 2.2. MIMO Technology

Multiple-input and multiple-output (MIMO) RADAR systems are used to enhance the data retrieved from hardware by creating virtual antenna arrays with multiple inputs and multiple outputs. The utilization of MIMO technology decreases the range of clustered detection. As the number of receive antennas increases, the range of clustered detection decreases. To explain further, each additional receive antenna, that is set a half wavelength apart from the previous, allows for the calculation of the extra distance covered by an IQ signal. The distance vector is derived from the extra length that each IQ signal must travel to reach an additional receiver antenna. The creation of an array of multiple receive antennas allows for the enhancement of the angular resolution. An FFT is used to exploit the relationship between the spatial resolution and the angle of arrival (AoA). MIMO technology is the utilization of virtual arrays consisting of multiple transmit and receive antennas to eliminate the need for additional antennas. Therefore, it enables the continuation of RADAR to be a lost-cost alternative when compared to other AV sensors [8].

### 2.3. Phase-Modulated Continuous-Wave RADARs

PMCW radar systems are designed to modulate a given signal’s phase. Coding schemes can be applied to control when the phase switches. Commonly, this is achieved by modulating the waveform in binary sequences. The binary sequencing allows for the transmitted radar signals to be orthogonal. Almost perfect auto-correlation sequence (APAS) codes are binary symbols mapped onto 0- and 180-degree phase shifts to incorporate a signal customized fingerprint [9]. After utilizing an APAS code, that customized fingerprint is what resolves mutual interference. By incorporating APAS codes using a high-speed analog-to-digital converter (ADC), PMCW radars can be fully functional no matter the distance between them. The only requirement is that each has its own APAS code, which can span from 1 bit to 1000 bits. Moreover, the incorporation of these binary phase shifts provides a solution to mutual interference issues that FMCW radars are not able to get around. PMCW radars have recently been able to be implemented on CMOS chips. The single-chip implementation allows for PMCW to be a feasible option for resolving mutual interference as well. Recent research in Germany showed that a 400 bit APAS code can resolve mutual interference when implemented in a PMCW system [10].

In the near future, most vehicles will be fully electric, with most having fully autonomous capabilities. The time between then and now will be shortened by advances in sensor development and signal processing, as mutual interference can be resolved by utilizing APAS codes with PMCW RADARs [10]. Currently, MFSK radars produce a fixed frequency when shifting between the step values. The incorporation of increasing and decreasing frequencies instead of a fixed frequency provides more data used for creating detection. The application of a signal fingerprint can be possible by allowing the frequency to be modulated in multiple different triangular chirps with different rates of modulating frequency.

## 3. Overview of Proposed Waveform

This section will summarize the effects of the proposed waveform, such as MFSK radar calculations, aspects of proposed waveform, and future of mutual interference.

### 3.1. MFSK Radar Calculations

MFSK radars are similar to standard FMCW radars. Each radar system sweeps a frequency in order calculate the radial velocity. IQ signals can be utilized in both systems to receive the angle of arrival (AoA) of an object. The difference between MFSK and FMCW radars is how they modulate their frequency. MFSK radars modulate their frequency in steps over a given frequency range. During each step, frequency does not modulate, as the MFSK radar progresses through each step and outputs a fixed frequency. Then, a fixed frequency is derived from the desired number of steps. The number of steps dictates what fixed frequency will be used during each step. Figure 6 shows one transmit signal of an MFSK waveform.

MFSK RADARs, with a step value of $10^{9}$, can be mathematically represented in Equation (Equation 3):(3)$$F_{step}=f_{s}+10^{9}\;\;for\;t_{n-1}\leq t<t_{n}\;\;;\;f_{s}=\text{76–81}\;\mathrm{GHz}$$

Binary frequency shift keying (BFSK) has been utilized in past Bluetooth communications and other wireless radio systems to implement a similar fingerprint aspect to that of PMCW radars. When comparing receiver (Rx) and transmitter (Tx) signals, beat frequencies can be calculated only if the frequency is modulated. When comparing these signals, it can clearly be shown that each transmit signal can both horizontally and vertically shifted. The vectors between each signal provide time delay information and Doppler data when using IQ signals. A FFT is used to convert data collected in the time domain by radars to the frequency domain. The conversion of this data allows for substantial amounts of data to be processed at a remarkably high rate. These data are stored, filtered, and then used to implement safe driving decisions. Each oscillation of a Rx signal provides a snapshot of the environment when compared to the Tx signal. In theory, an AV would perform best if it could capture images of the environment at a rate that is inconceivably fast. Accomplishing an extremely fast refresh rate would allow an AV to make decisions in slow motion compared to the time system that humans are used to. The amount of data lost is directly proportional to the speed of the vehicle. In fact, every sensor on any creation is limited to the rate at which it collects data. Thanks to advances in recent years, higher-speed data processing has become quite feasible considering the number of calculations needed for AVs and where we were two decades ago [11].

### 3.2. Aspects of Proposed Waveform

The proposed waveform is a variation of a MFSK radar. Instead of outputting a fixed frequency during each step, we propose the waveform modulated during each step positively and negatively in a linear fashion. An up-chirp followed by a down-chirp creates a triangular modulation pattern during each step. This pattern allows for beat frequencies to be calculated in the same fashion as FMCW and MFSK radars. As a RADAR beam is transmitted and received, when plotted with each other, received signals are shifted. The horizontal and vertical distances show the difference in domain and range values correlated to the direction and velocity of an object. Each shift, depending on the location relative to the Tx signal, produces some variation of a combined x and y vector that can represent both shifts with the angle relative to the time axis. Additional triangular sweeps can be implemented to collect more beat frequencies, which help eliminate ghost detection [12]. As the number of beat frequency calculations increases, the number of vectors that can be drawn through and confirmed with real detection increases. As computations are added to modify the waveform, the entire process is computationally more expensive. Each modification of our proposed waveform’s variations allows for the number of rearrangements to increase proportionally to the factorial numbers. The sensitivity of the antenna array would dictate the number of steps and linear modulation patterns that could be implemented. Along with this, the angular resolution for any continuous-wave RADAR is proportional to the number of received antennas. The number of individual signals that can be created would increase as the numbers of different steps and frequency slopes are increased. The proposed five steps and three different slopes allow for 720 different signals to be created. These 720 different signals are the result of 3! × 5! = 6 × 120 = 720. If the proposed radar had the ability to increase either the number of steps or the number of different slopes it could detect, the number of individual signals would increase factorially. The number of binary digits used to control the sequencing of each would increase as well, which would increase the computational power needed.

### 3.3. Future of Mutual Interference

Mutual interference happens when Rx antennas are blinded by Tx antennas on another vehicle using the same frequency range. Therefore, it becomes a widespread problem when mass producing AVs. Recent PMCW research has shown that it can be resolved using binary symbols. APAS codes have shown the capability of producing a remarkably high number of unique signals to be used for each individual vehicle on the road [9]. We propose similar methods that can be applied to current MFSK radars. One of the first steps for developing a coding scheme for a signal is to decide how many different devices will be used in the assigned frequency band. Considering, in recent years, the success that auto manufacturers have had in terms of AVs, these vehicles will one day be a consumer standard. The elimination and prohibition of gasoline vehicles are right around the corner. Taking all that information in, it was only reasonable to calculate the potential number of cars during a high-traffic scenario. The number of signals needed was calculated using a bidirectional long-range RADAR (LRR) of 600 m. A 60 m 10-lane highway was used to illustrate the area of the road covered by LRRs and the area of a sedan electric car. It was found that 3275 individual signals would have to be created to ensure no car was emitting the same signal.

## 4. Coding Schemes

After the effects of the proposed waveform and mutual interference, a random coding scheme and the results of the proposed waveform in a quantitative manner are presented.

### 4.1. Random Coding Scheme

PMCW offers key insight into the benefits of applying a coding scheme to an automotive radar. High-speed ADCs enable the coding of binary symbols into PMCW waveforms. Many other types of electronics, primarily sensors, are programmed in the same manner. Controlling the order in which each step is taken allows for binary coding to be implemented. MFSK radars tend to sweep linearly in steps; the step order can be rearranged to allow different frequency orders to be implemented. We proposed that each step modulates over a 1 GHz range. If the range it sweeps is from 76 GHz to 80 GHz, the MFSK would only take five steps to sweep the entire range. If the MFSK only takes five steps, then that means the radar has 120 different step rearrangements. Binary coding can be used with six variables to control the output of one given rearrangement. Since the number of variations can be represented by 5!, six binary digits in total must be used to control the signal. One combination of a binary encoded MFSK signal is shown in Figure 7.

This type of coding scheme is proportional to the number of steps taken, assuming that the specified frequency band is fixed. Binary coding will not work if applied because the fixed frequency band limits the number of steps. Since the frequency band is considered fixed, the implementation of many steps leads to the signal becoming indistinguishable from noise. As the number of steps increases, the chirp time decreases as well as the domain values for each triangular pattern.

### 4.2. Proposed Waveform

The proposed waveform is divided into five parts: (1) the importance of coding steps and modulation patterns, (2) signal parameters, (3) multi-object tracking results, (4) mutual interference results and (5) advantages and disadvantages.

#### 4.2.1. The Importance of Coding Steps and Modulation Patterns

Using MFSK, creating more steps allows for more binary digits to be implemented at the cost of the degradation of the signal. The implementation of multiple coding schemes is the key to creating a higher number of signal combinations for MFSK radars. If the frequency during each step is modulated, triangular modulation would be possible. Controlling how frequency is modulated through each step is vital to the creation of unique signals. An up-chirp followed by a down-chirp produces the outline of a triangle on a spectrogram using a FFT data. The number of different triangles produced allows for more ghost detection to be eliminated by comparing the time delay and the beat frequencies of the Rx and Tx antennas. The introduction of an MFSK radar that creates two differently sloped triangles allows for eight beat frequencies to be calculated, lowering the false-alarm rate for ghost detection. In this case, the slope of the triangle is proportional to frequency over time. Triangular modulation is accomplished by changing the rate at which frequency increases or decreases. It is beneficial to modulate the frequency of an up-chirp and down-chirp with the same rate of change. Incorporating this feature into a signal enables time delays to cancel, allowing for a more stable signal to be generated. With a smaller angle between the chirps, the signal would be less stable. Along with decreasing the probability of ghost detections, this method of signal propagation allows for many types of coding schemes to be implemented. The previous binary scheme offers insight into the implementation of multiple coding schemes. The number of steps should be equivalent to the number of channels. As the number of steps increases, assuming that the frequency range is fixed, the bandwidth of the channel decreases.

#### 4.2.2. Signal Parameters

By introducing different sequences of chirps with different rates of frequency change, several different chirp sequences can be compared to one another. MFSK radars have three different customized properties to introduce a higher number of unique signals: (1) the utilization of different channels, (2) changing the rate at which frequencies modulate, and (3) the incorporation of multiple differently sloped chirp sequences. These three properties are the basis on which the proposed coding scheme is derived. All three customized properties are shown in Figure 8.

The proposed triangular waveform can be mathematically represented in Equation (Equation 4):(4)$$F_{triangular}=\left(\pm \left(m_{x}t\right)+f_{s}\right)\;\;for\;\;t_{n-1}\leq t<t_{n}$$
where *f${}_{s}$* is the y-intercept at various frequencies, *m${}_{x}$* is the slope at 0.25, 0.5, and 1, and *t* is the x-intercept.

The coded variables in the proposed triangular FMCW signal are: A = 1 and B = 2 for the slope ordering; and C = 1, D = 2, E = 3, and F = 4 for the step ordering. A–F are coded variables used to control the ordering of both slopes and steps. These variables must be included in order to effectively produce any combination of the proposed waveform. In addition, the combination of proposed coding schemes enables the number of possible signal to increase factorially. A MATLAB coded plot of frequency modulation is presented in Listing 1, where the first point is a function of the waveform’s chirp time.

**Listing 1.** Waveform Point Generator.*% Initialize*p1x = 0;deltaY = 1;*%----------Step Parameters----------%*firstStep = 76;secondStep = 77;thirdStep =78;fourthStep = 79;fifthStep = 80;divisionsHz = 5; *% total number of steps /  channels**%----------Slope Parameters----------%*firstSlope = .25;secondSlope = .5;thirdSlope = 1;divisionsSlope = 3; *% total number of slopes* *% Sweep number of points*
delY = [ firstStep ,secondStep ,thirdStep ,fourthStep ,fifthStep ];
Slope = [ firstSlope ,secondSlope ,thirdSlope ];
n = divisionsHz;

*% Generate points from nested loop*
       **for** ct = 1:n
             **for** mct = 1:divisionsSlope               m = Slope(mct);               **hold** on               p1y = delY(ct);               p2y = p1y + deltaY;               p3y = p1y;               p2x = (p2y–p1y)/m +  p1x;
               width = 2*(p2x–p1x);               p3x = p1x + width;               xup = [p1x p2x];               yup = [p1y p2y];               xdown = [p3x p2x];               ydown = [p3y p2y];               p1x = p3x;
      **plot**(xup, yup, ’–k’, LineWidth=2)
      **plot**(xdown , ydown, ’–k’, LineWidth=2)            **end**
      **end**

#### 4.2.3. Multi-Object Tracking Results

The solid Tx signal and dotted Rx signals in Figure 9 represent three potential objects being tracked. These objects can be detected by calculating the beat frequency during periods of modulation. Beat frequencies can be calculated using the same math values as FMCW and MFSK waveforms.

To calculate the Doppler and time delay values in the same manner, the Tx and Rx signals must be mixed and filtered to output a beat frequency. This beat frequency can be plotted with other beat frequencies on the same plot. Figure 10 represents a plot of all six beat frequencies for each of the three objects. These calculations result in eighteen different beat frequencies per chirp period, rather than just six beat frequencies, when tracking three objects with FMCW. The increase in beat frequencies validates the proposed waveform by visually showing better detections. The coordinates provided are the magnitudes of Rx signal delay values.

The rates of modulation enable the tracking of three objects during one chirp sequence. An Rx signal may have Doppler delay, time delay, both, or neither. Therefore, the received signal may be translated horizontally to the right to indicate time delay, or vertically translated to indicate Doppler delay. The true location of an object may be found by calculating the beat frequency. The beat frequency is the product of transmitted and received signals after it has been low-pass filtered. With a standard FMCW RADAR, there are two possible beat frequency calculations due to the triangular frequency modulation. During the up and down sweeps, the proposed waveform includes three modulations, obtaining differently sloped beat frequencies. As seen in Figure 5, two ghost detections appear due to the same rate of modulation being used. When using different modulations rates, the slopes of the beat frequencies change proportionally to the rate of frequency modulation. Each of the three up-sweeps provides three positively sloped beat frequencies, likewise for the down-sweeps. Each chirp sequence of the proposed waveform offers six beat frequency calculations to enable the tracking of three objects at a time. Moreover, eighteen unique vectors are shown in Figure 11.

#### 4.2.4. Mutual Interference Results

The driving scenarios shown in Figure 2 explain how sensors may cause mutual interference when outputting Tx signals. The proposed waveform was designed to resolve mutual interference through modulating in different frequency bands. As seen in Figure 12, two Tx signals are generated using the protocol code presented in Listing 1. During periods in which the two Tx signals are outputting in different frequency bands, there is no possibility of mutual interference.

#### 4.2.5. Advantages and Disadvantages

One of the advantages is that beat frequencies are calculated using the same mathematical approach utilized for both FMCW and MFSK RADARs. Thus, it is still feasible to use Doppler and time delay to compute distance. Additionally, the suggested waveform produces up to ninety beat frequencies every period. Frequency modulation rates and running protocol code may produce unique signals that resolve mutual interference. Since the proposed waveform outputs three triangles at five frequency values, it will operate in short periods when undergoing mutual interference. Five starting frequencies for a chirp enable the chirp to operate on a frequency, even when presented as in Figure 2. Table 1 shows aspects of the continuous-wave RADAR waveform. A disadvantage of implementing the proposed coding scheme is the high computational costs associated with creating and monitoring new data points.

## 5. Conclusions

This paper dove into the aspects of previous and current progress pertaining to radar waveform. Research works on FMCW, PMCW, MIMO, and MFSK radar systems have each shown extraordinary examples of pushing the limits of signal processing. The proposed waveform is a type of MFSK radar that uses code to resolve mutual interference. Currently, MFSK radars step and output a fixed frequency and continue to the next step with a higher fixed frequency. This paper proposed a MFSK waveform that modulates frequency during steps. During each step, multiple chirp sequences are implemented to increase the number of beat frequency calculations. The number of beat frequency calculations enables more Doppler and time delay data to be processed. Along with increasing beat frequency calculations, chirp sequences are modulated at different rates to increase the number of unique signals for multi-object tracking. The total number of signals increases factorially as more slopes or steps are introduced. The proposed waveform offers a resolution of mutual interference by introducing more types of signals than what is used on today’s AV RADARS.

## 6. Future Work

The implementation of the proposed waveform is coded and functional. The real-world implementation of this waveform may require a unique antenna array, most likely using MIMO technology to maintain low hardware cost. Simulations of the waveform are performed on MATLAB. The proposed waveform will be programmed to generate radar data based off the beat frequencies created previously. The proposed coding scheme has future potential by increasing the amount of slopes or steps. The incorporation of MIMO technology may be a viable step for creating more signals. With that, ghost detections may cancel using virtual antenna arrays. As technology evolves, AVs will encounter a decentralized tracking network, in which detections will be shared amongst vehicles.

## Figures and Tables

**Figure 1 sensors-23-07192-f001:**
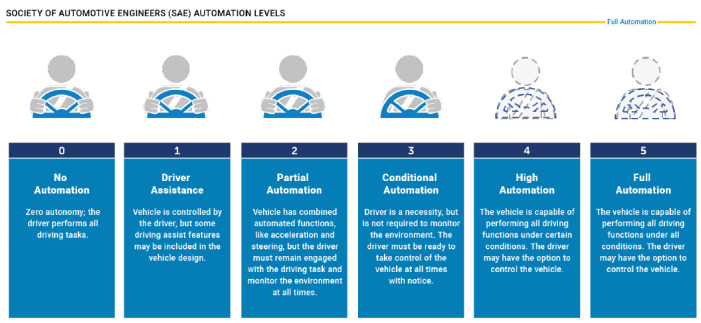
Society of Automotive Engineers automation levels.

**Figure 2 sensors-23-07192-f002:**
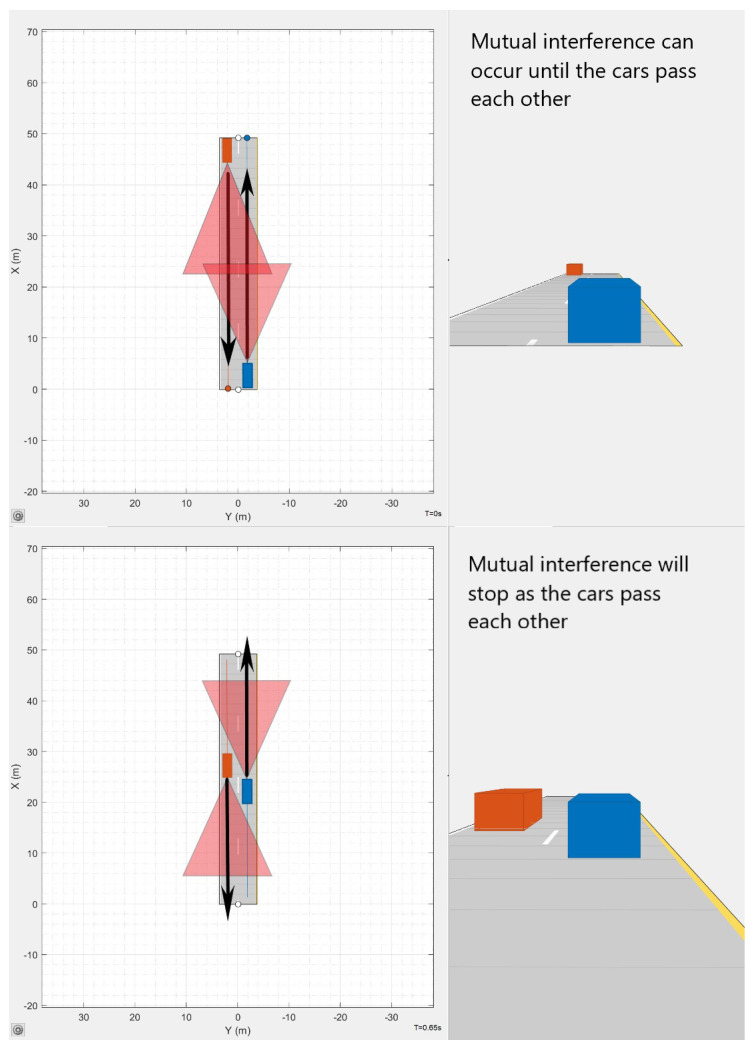
Mutual interference when vehicles approach each other.

**Figure 3 sensors-23-07192-f003:**
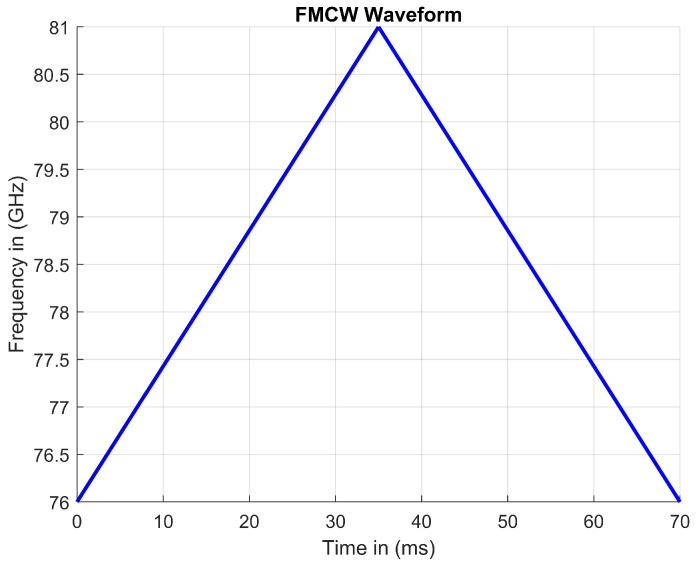
A standard FMCW signal over a specified frequency range in a given period of time.

**Figure 4 sensors-23-07192-f004:**
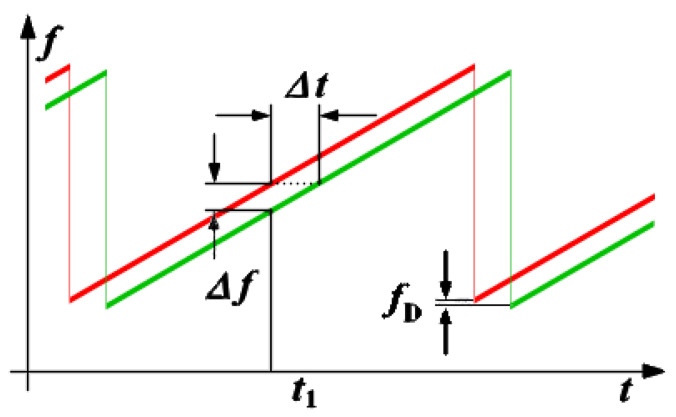
Doppler and time delay of a rising and falling edge signal [5].

**Figure 5 sensors-23-07192-f005:**
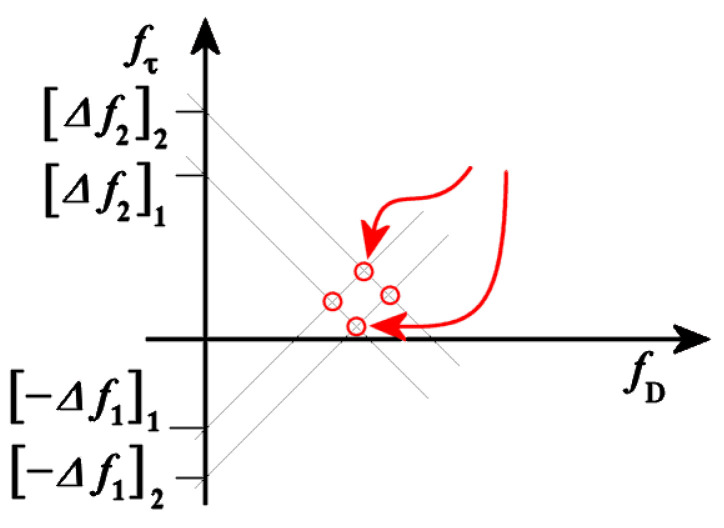
An object and ghost detention circle indicator signal [5].

**Figure 6 sensors-23-07192-f006:**
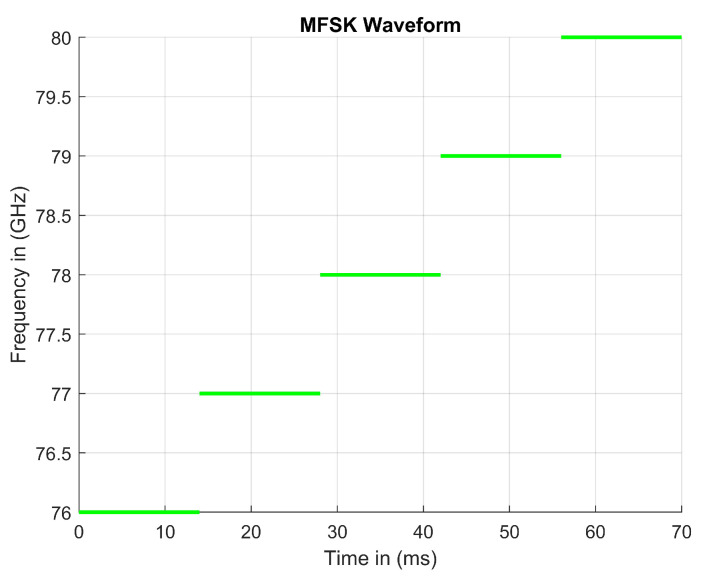
Standard MFSK signal steps with a fixed frequency during a given time period.

**Figure 7 sensors-23-07192-f007:**
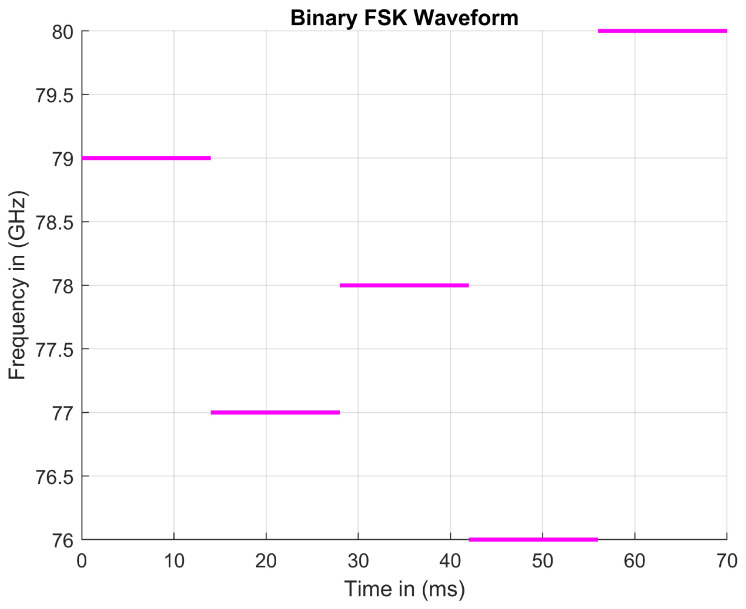
A random standard FMCW signal over a specified frequency range in a given period of time.

**Figure 8 sensors-23-07192-f008:**
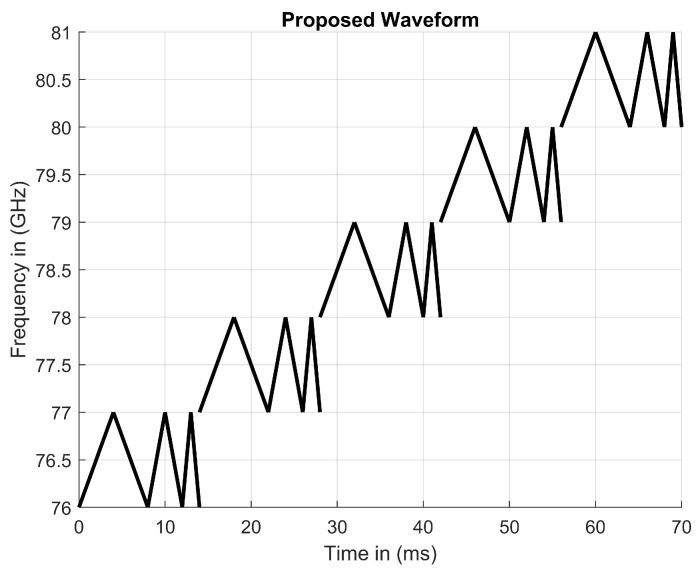
A proposed triangular FMCW signal over a specified frequency range in a given period of time.

**Figure 9 sensors-23-07192-f009:**
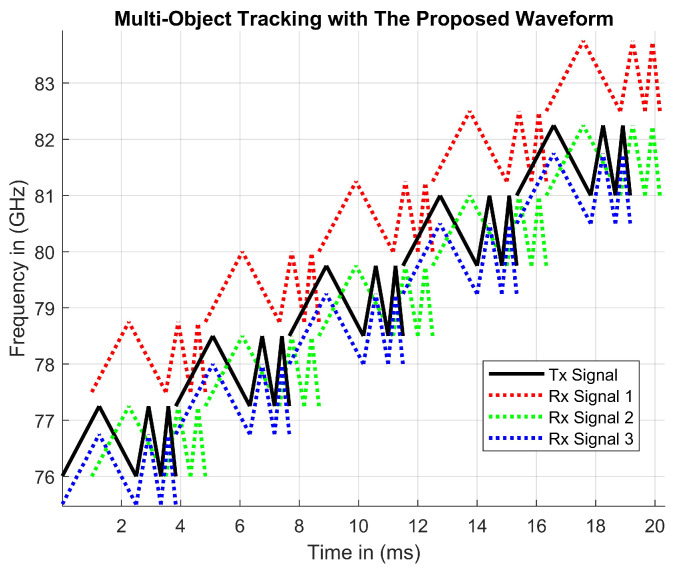
A transmitter and multiple receiver signals from hypothetical objects.

**Figure 10 sensors-23-07192-f010:**
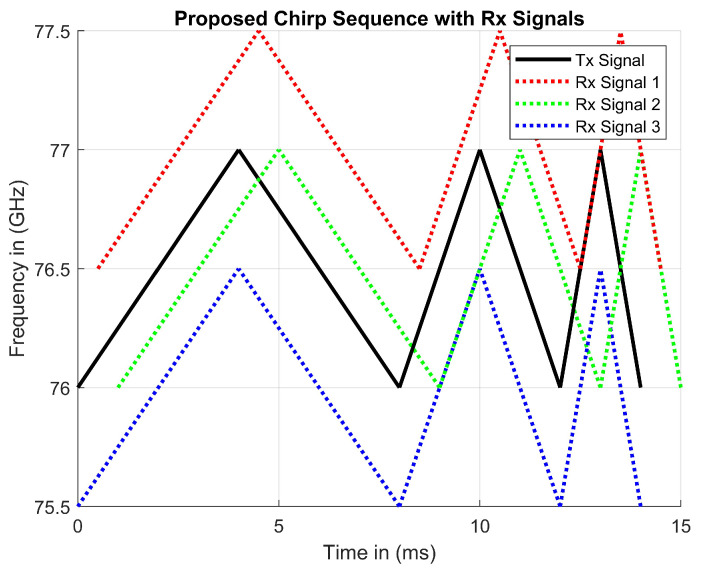
Three objects being tracked within one chirp sequence.

**Figure 11 sensors-23-07192-f011:**
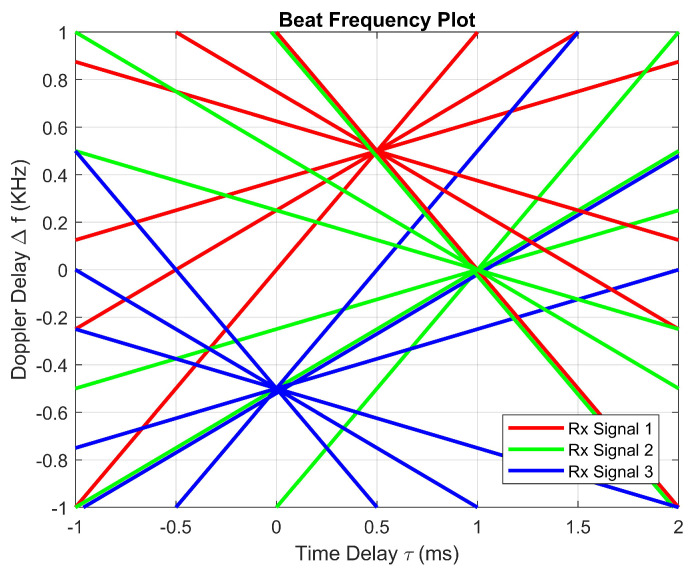
Three object detections seen at coordinates (0,−0.5), (0.5,0.5), and (1,0).

**Figure 12 sensors-23-07192-f012:**
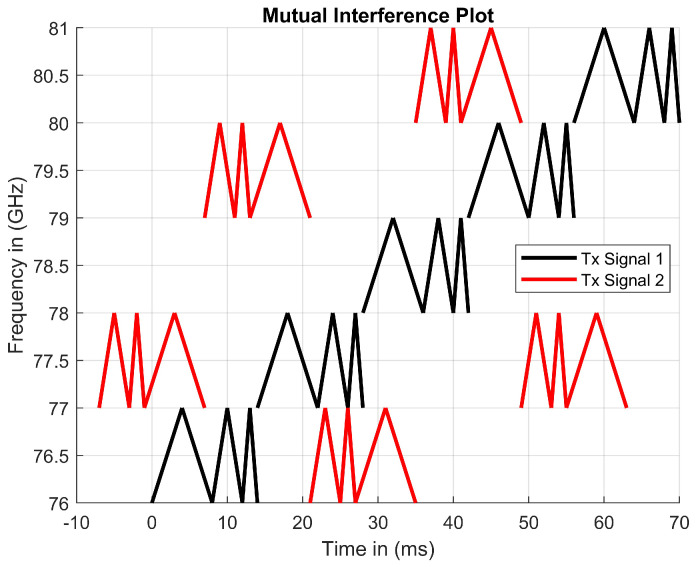
Two generated Tx signals in different frequency bands.

**Table 1 sensors-23-07192-t001:** Comparisons between continuous-wave radars.

RADAR Waveforms	Beat Frequency Calculations per Period	Pattern	Description
Frequency Modulated	1-2 per received signal	Sawtooth or Triangular	Modulates the entire frequency band in a linear fashion, positively or negatively
Phase Modulated	N/A	Often looks like a broken sinusoidal wave due to phase shifts interrupting the signal	A code is generated to map phase shifts onto the signal. The entire period of the signal is required to compare Tx and Rx signals
M-ary Frequency Shift Keying	Proportional to the number of steps	Staircase with no vertical vectors	Offers a method for increasing the number of beat frequencies, which enables multi-object tracking
Binary Frequency Shift Keying	Proportional to the number of steps	Random horizontal vectors	A code generates the ordering of frequency values
Proposed Waveform	6 per received signal	5 random sets of 3 differently sloped triangles	Offers more beat frequency calculations than MFSK

## Data Availability

Not applicable.

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
