# Peer review of "A Modulated Approach for Improving MFSK RADARS to Resolve Mutual Interference on Autonomous Vehicles (AVs)"

_sensors, 2023, doi:10.3390/s23167192_

Round 1

Reviewer 1 Report (Previous Reviewer 1)

Dear authors,

Thank you for carefully revising your paper based on the reviewers' feedback. I have thoroughly reviewed your responses and modifications, and I commend you for your positive and diligent efforts in addressing each reviewer's suggestions. Here is my specific evaluation of your changes and responses:

Regarding the concerns raised by Reviewer 1 about Figure 2 and the representation of FMCW signals, you have clarified the issue and made necessary modifications to avoid visual confusion. Additionally, you have provided a reasonable explanation of the advantages of IQ signals in continuous wave RADAR systems. The addition of a table to organize and clarify the paper enhances its readability and accuracy.

Your response to Reviewer 2's concerns about chart visibility and the Real-Imaginary coordinate issue is commendable. By re-rendering the charts to improve visibility and revising the relevant section while removing Figure 3, you have effectively addressed the coordinate confusion. These modifications better convey your research findings.

I appreciate that you have taken Reviewer 3's suggestions to heart, resulting in clearer and more concise section divisions. The inclusion of a Pros and Cons section, gathering quantitative data, and removing Figure 9 while supplementing with more interpretable data have enhanced the paper's transparency and credibility.

Your response to Reviewer 4's feedback regarding providing specific comparisons, correcting spelling issues, and adding specific AV scenarios and components is commendable. By offering more concrete comparisons, rectifying spelling errors, and introducing real-world AV scenarios and components, you have improved the clarity and practical relevance of the paper.

Overall, you have diligently and positively addressed the reviewers' comments and suggestions, leading to significant improvements in the paper's quality and suitability for publication. However, I recommend conducting a final review before submission to ensure that all modifications have been accurately implemented and that there are no unresolved issues or potential areas for improvement. For example, “Mutual interference is more common in radars than ultrasonic devices as they operate in a narrower regions of space.[3].” Is there an extra period in this sentence?

Once again, thank you for submitting your paper to our journal. We look forward to reading the final version of your paper and hope that your research will make a valuable contribution to the academic community.

Best regards,

Author Response

Thank you for your comments. A final review of the paper has been conducted. Minor grammatical errors have been corrected.

Reviewer 2 Report (Previous Reviewer 2)

After conducting a comprehensive review of both the original manuscript submitted to Sensors and its subsequent revised version, I am delighted to highlight the remarkable enhancements that have been made. The modifications incorporated in the new iteration have undeniably captivated my attention and garnered my resounding approval. Thus, I am pleased to announce my acceptance of the revised manuscript and express my utmost satisfaction with the revisions implemented.

Author Response

Thank you for your comments. A final review has been conducted as a precautionary measure.

Reviewer 3 Report (Previous Reviewer 3)

Author Response

Thank you for your feedback. Please find the answers to your questions below.

1. The number of binary digits would increase in a factorial manner in relation to the steps. Ensuring each unique signal would have it's own set of binary digits. The number of these digits would increase if you increased the number of steps, and vice versa. The number of steps within one chirp period of the proposed waveform is limited by the FCC automotive radar band. MFSK RADARs can use a higher number of steps, we personally have seen around 30-40 steps over a 5 GHz range, so it can likely be increased from the proposed 5 steps.

2. AVs would need to communicate within a network to achieve this. To implement these unique signals, an API request would be needed to ensure no two cars on the road would be using the same signal. It would also be beneficial to ensure two cars within a certain range could operate on different frequencies. No algorithms have been used yet.

3. We picked those slopes to show a larger visible difference between the slopes of the triangle. If you were to increase the number of triangles with different slopes, you would be able to track more objects. Since we proposed 3 very different slopes, we are able to track 3 objects clearly. We also picked 3 due to the clustering that can occur when having a large amount of beat frequencies over a chirp period. If you were to have 5 slopes, you would be able to calculate 25 beat frequency vectors with our waveform, which ends up having visual difficulties. We have made changes in section 3.2 of the paper to clarify.

4. The math for continuous waveform RADARs has shown to be similar across both FMCW and MFSK waveforms. PMCW has shown to be calculated with the use of a 400+ bit APAS code, but mathematically the time and doppler delays should be calculated in the same manner. The benefits of doing so include tracking multiple objects at once and lower power output since the wave is continuous. We have added changes in section 2.1 of the paper to add emphasis.

5. We have not tested the waveform's range ambiguity yet, but it would easily be modifiable with our current coding scheme. Since we are able to change the number of triangles, steps, and slope values in our code, finding an optimal chirp time would be a straight forward process.

6. We have not looked too far into this yet, but assume another onboard processor would be required to run computations. The antenna array should not have to change if the RADAR were using the same wavelength. Equation 2 in our paper shows the relation between receive antennas and wavelength. An ADC could be used to communicate the modulation pattern. We have made changes in section 3.2 of the paper to include more information about potential implementation.

7. Standard MFSK RADARs are prone to tracking multiple objects with similar distances as they do not modulate during their steps. Our proposed waveform was designed to get around this problem. A metric that could be used to further verify this would be to test the performance of multi-object tracking in a simulated, then real environment. This metric would show how frequency modulation during steps increases detection quality.

8. The expected impact has not been fully computed yet, but we assume the steps and different modulation rates would require upwards of twice the computation level required to run current FMCW RADARs.

9. The code can be easily modified to include a higher number of triangles or steps. Increasing either of these would increase the total number of signals in a factorial fashion. We have made changes in section 4.2.2 of the paper to emphasize.

10. The direction of objects in the environment would be determined by IQ signals. Our proposed waveform was intended to modulate signals during steps. However, tracking multiple objects traveling at similar velocities in the same direction hasn't been evaluated yet due to its possible visual complexity. Assuming the antenna array had enough receive antennas, the resolution should be well enough to distinguish various objects in different lanes. With less receive antennas, it's likely to see a wider vehicle.

A final review has been conducted.

Reviewer 4 Report (Previous Reviewer 4)

The author responded well to my comments. After carefully modifying the format, it can be accepted

The author responded well to my comments. After carefully modifying the format, it can be accepted

Author Response

Thank you. A final review has been conducted as a precautionary measure.

This manuscript is a resubmission of an earlier submission. The following is a list of the peer review reports and author responses from that submission.

Round 1

Reviewer 1 Report

It is suggested to revise the manuscript to better justify the study design, as follows.

1)  Please explain“Is the signal shown in Figure 2 the transmitted signal or the received signal?” Please explain why there is a time difference between two signals and what is the significance of this time difference?

2)  Please briefly explain the meaning of ghost detection and why the two red circles in Figure 5 are called ghost detection?

3)  This study compared the proposed method with other frequency method systems. It is suggested to present a concise summary and comparison in the form of a table, which would provide a clear and organized overview.

Author Response

Thank you for taking the time to read and review our paper, your feedback is greatly appreciated.

  1. Figure 2. can be referenced as a transmitted signal. Multiple objects would provide multiple received signals. It shows modulation periods for a FMCW signal, which modulates through a specified frequency band. This is not what FMCW looks like in real life. Each signal graph in the paper has been modified to not show the output of IQ signals due to possible visual confusion. In reality, all continuous wave RADAR systems should output two of the same signal 90 degrees out of phase.
  2. Ghost detections are false readings given by a device collecting information about the environment, common devices that output and receive waves. Due to the math involved in continuous wave RADAR systems, it is beneficial to output IQ signals to collect data from the received signal. When a signal outputs a triangular wave, the signal will provide two beat frequencies per chirp. The higher number of beat frequencies enables more objects to be tracked.
  3. I agree that a table will help organize and clarify the paper. Therefore, a table will be added to the revised version.

Again thank you for your feedback, all of your comments have been addressed.

Reviewer 2 Report

Original Submission

Recommendation

Rejection

Comments to Author:

Title:  

A Modulated Approach for Improving MFSK RADARS to Resolve Mutual Interference on Autonomous Vehicles (Avs(

Overview and general recommendation.

This paper explores the utilization of RADAR techniques and their significant applications, focusing on addressing mutual interference in autonomous vehicles. While I appreciate the author's expertise in LIDAR/SAR/RADAR and find the topic intriguing, I must provide an honest assessment of the paper. The abstract adequately summarizes the idea and concepts, although it would benefit from a native English speaker's review. Additionally, the introduction effectively sets the stage for the presented research. However, there are areas that require improvement, such as including important references and enhancing the quality of the figures. The results, discussions, and conclusions sections are satisfactory.

Detailed comments:

1.    Figure 1: It is crucial to ensure that the figure does not infringe on any copyrights. Additionally, the quality of all figures is very poor. It is recommended to use high-quality figures in the *.eps format.

2.    Eq.2: Consider removing this equation as it may not be necessary for the paper's main focus.

3.    Figure 3: The figure appears to be irrelevant and does not contribute effectively to the paper. Please consider removing it.

I have reservations about accepting this paper in its current state. Although the experiments show promise, substantial improvements are required. Failure to address the mentioned issues would lead to rejection. Additionally, the paper appears to lack substantial content, potentially falling below the standards of MDPI. I suggest considering a more suitable publication venue with a maximum impact factor of 1.5 for this paper.

Please revise and address the points mentioned above to enhance the quality of your work.

Top of Form

English IS OK!

Author Response

Thank you for taking the time to read and review our paper, your feedback is greatly appreciated.

1. We will re-render the majority of the figures due to visibility issues.

2. We realize that the Real and Imaginary axes could lead to confusion. With that being said, that section has been revised and figure 3 was removed.

Again thank you for your feedback, all of your comments have been addressed.

Author Response

Thank you for taking the time to read and review our paper, your feedback is greatly appreciated.

  1. We agree that the paper should be further divided into clear and concise sections. This is fantastic feedback and will surely add value to our paper.
  2. Pros and Cons have been included in the paper (sec. 4.2.5)
  3. We worked on gathering quantitative data as shown in section 4.2.3 and 4.2.4 of the revised paper. 
  4. Figure 9 has been removed as more interpretable data was added.

Again thank you for your feedback, all of your comments have been addressed.

Reviewer 4 Report

This manuscript discusses a automotive radar waveform involving the theory behind of Mary Frequency Shift Key radar systems. A solution to mutual interference is provided along with the Mary Frequency Shift Key theory. The topic of this paper is interesting. However, there are some issues that need to be properly addressed.

1. The current progress pertaining to radar waveform should be more widely elaborated. More specific comparisons should be presented to highlight the difficulties of this manuscript.

2. “M-ary Frequency Shift Key” in the abstract should be revised as “Mary Frequency Shift Key”. More spelling mistakes should be checked and revised.

3. The waveform proposed in this manuscript is suggested to be analyzed in conjunction with specific application scenarios and components of autonomous vehicles.

Suggest checking and modifying the spelling of the entire text

Author Response

Thank you for taking the time to read and review our paper, your feedback is greatly appreciated. Revisions have been made thoughout the paper including your comments and suggestions.

  1. More concrete comparisons have been provided to the audience to make it clear and consice.
  2. The spelling has been revised thoughtout the paper.
  3. Specific AV scenarios and components have been added.

Again thank you for your feedback, all of your comments have been addressed.